# Implementation of GeneXpert MTB/Rif proficiency testing program: A Case of the Uganda national tuberculosis reference laboratory/supranational reference laboratory

Joel Kabugo[1]*, Joanita Namutebi[1], Dennis Mujuni[1,2], Andrew Nsawotebba[1,3], George William Kasule[1], Kenneth Musisi[1], Edgar Kigozi[2], Abdunoor Nyombi[1], Pius Lutaaya[1], Fredrick Kangave[1], Moses L. Joloba[1,2]

1 Uganda National Tuberculosis Reference Laboratory, Kampala, Uganda, 2 Makerere University College of Health Sciences, Kampala, Uganda, 3 Uganda National Health and Laboratory Diagnostic Services, Kampala, Uganda

* ksolomonjoel@gmail.com

**Data Availability Statement:** All relevant data are within the paper and its Supporting Information files.

## Abstract

### Background

Following the WHO's endorsement of GeneXpert MTB/RIF assay for tuberculosis diagnosis in 2010, Uganda's ministry of health introduced the assay in its laboratory network in 2012. However, assessing the quality of the result produced from this technique is one of its major implementation challenges. To bridge this gap, the National tuberculosis reference laboratory (NTRL) introduced the GeneXpert MTB/RIF proficiency testing (PT) Scheme in 2015.

### Methods

A descriptive cross-sectional study on the GeneXpert PT scheme in Uganda was conducted between 2015 and 2018. Sets of panels each comprising four 1ml cryovial liquid samples were sent out to enrolled participants at preset testing periods. The laboratories' testing accuracies were assessed by comparing their reported results to the expected and participants' consensus results. Percentage scores were assigned and feedback reports were sent back to laboratories. Follow up of sites with unsatisfactory results was done through "on and off-site support". Concurrently, standardization of standard operating procedures (SOPs) and practices to the requirements of the International Organization for Standardization (ISO) 17043:2010 was pursued.

### Results

Participants gradually increased during the program from 56 in the pilot study to 148 in Round 4 (2018). Continual participation of a particular laboratory yielded an odd of 2.5 [95% confidence interval (CI), 1.22 to 4.34] times greater for achieving a score of above 80% with each new round it participated. The "on and off-site" support supervision documented

**Funding:** The author(s) received no special funding for this work. The financial support to run all the above activities was part of the routine funding through the Regional Global fund through the East, Central and Southern Africa (ECSA) Health Community Project to the Supra-National Reference Laboratory of Uganda to support Tuberculosis (TB) diagnostic implementation and quality management systems in the Laboratories performing TB testing.

**Competing interests:** The authors have declared that no competing interests exist.

improved performance of failing laboratories. Records of GeneXpert MTB/RIF PT were used to achieve accreditation to ISO 17043:2010 in 2018.

## Conclusion

Continued participation in GeneXpert MTB/RIF PT improves testing accuracy of laboratories. Effective implementation of this scheme requires competent human resources, facility and equipment, functional quality management system, and adherence to ISO 17043:2010.

## Introduction

The GeneXpert MTB/RIF assay since its recommendation for tuberculosis (TB) diagnosis in 2010 by WHO has been used worldwide. In Uganda, the assay was introduced in the laboratory network in 2012 to universalize access to TB services including notification, treatment, and prevention [1, 2]. The potential benefits of the assay concerning patient management are more noticeable at the lower healthcare facilities of a country's healthcare system than at the national level and reference testing settings. This is because lower healthcare facilities are more closer to patients which reduces the turnaround time (TAT) to report *Mycobacterium tuberculsosis* (*M.tb*) diagnosis results. The challenge comes with verification of the quality of the results produced and since lower healthcare facilities such as health center three (HCIII) have less developed quality management systems, this may be compromised [3, 4].

The most effective and credible way to obtain accurate, reliable, and cost-effective results is through the implementation of a laboratory quality management system (LQMS) [5]. The healthcare system in developing countries has immeasurably recognized the impacts of LQMS when it comes to patient care and is gradually applying it [6]. It has indeed become necessary for all countries to strengthen the capacity of clinical laboratories and this can be achieved through setting up systems that monitor performance such as external quality assurance (EQA) [7, 8]. Recent studies in Africa show that the laboratory fraternity and test quality for all types of clinical laboratories remain poorly established [9, 10]. EQA in TB performing laboratories is implemented through; onsite support supervision, blinded rechecking, and proficiency testing (PT) [11].

With GeneXpert MTB/RIF test, blinded rechecking/retesting is not possible since the tested samples are discarded immediately after testing. This puts PT as the best-suited technique to assess the performance of laboratories with the method involving sending known samples to participants [8]. PT intends to assess all phases of diagnostic testing from pre-analytical, analytical and post-analytical. It involves periodically sending multiple well-characterized strains to participants for analysis. Each laboratory's result is compared with the expected result and the consensus of other participants. Testing sites with discordant results and errors are identified, timely feedback is sent including recommendations for corrective action, and where necessary on-site supervisory visits are conducted to address nonconformities[12, 13].

National TB programs in most resource-limited countries especially in Sub-Saharan Africa have not been able to establish PT in the GeneXpert MTB/RIF assay [14]. This hardship is mainly attributed to many aspects which include: technical incompetence to prepare panels, lack of infrastructure, lack of human resource, high cost to procure PT preparation equipment, lack of suitable means of transporting panels to laboratories, supply chain management breakages, poor connectivity and Information Technology (IT) [4, 15, 16]. Implementation of EQA

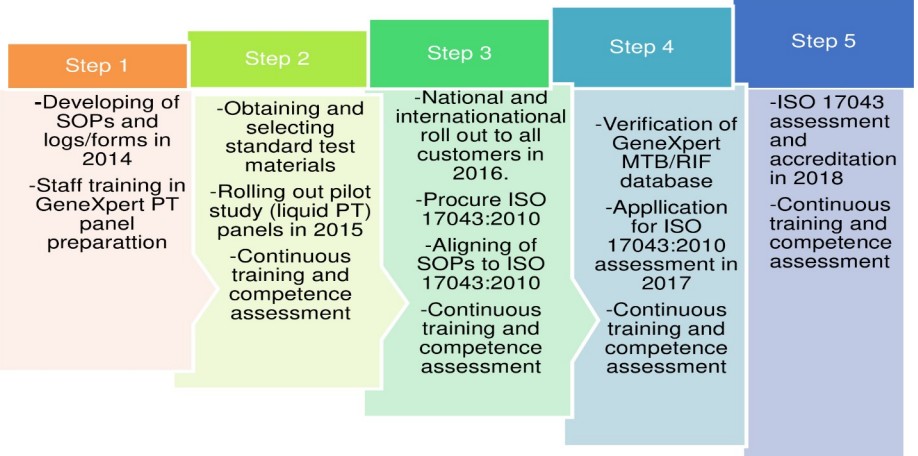

**Fig 1. Roadmap for establishing GeneXpert PT scheme in Uganda.**

through organizing regular PT rounds and identification of training needs is one of the key aspects for efficient tuberculosis (TB) laboratory network [17].

According to a recommendation from the World Health Organization's Regional Office for Africa (WHO AFRO), National public health reference laboratories have to develop and implement a quality management system (QMS), which encompasses the provision of PT services to all sites under the network [9, 18]. As a result, the Uganda national TB reference laboratory (NTRL) [19] which is also a World Health Organization (WHO) supranational TB reference laboratory (SNRL) designed a GeneXpert MTB/RIF PT Scheme to promote the quality of TB diagnosis when using the GeneXpert MTB/RIF platform. This study was conducted to demonstrate the lessons and attributes of scaling up the implementation of GeneXpert PT in Uganda.

## Materials and methods

Uganda NTRL conducted a descriptive cross-sectional study from October 2015 to May 2018 among GeneXpert MTB/RIF testing laboratories. In this study, standard test samples were sent out to enrolled testing laboratories within and out of Uganda to assess the quality of *Mycobacteria tuberculosis* complex diagnosis using the GeneXpert MTB/RIF® platform.

An implementation road map was drawn and activities generated as shown in "**Fig 1**".

### GeneXpert MTB/RIF PT scheme concept inception and principle

The Uganda NTRL and a few of the peripheral laboratories it supports started the use of GeneXpert MTB/RIF® technology as a TB diagnostic method in 2012. The laboratory scientists at NTRL received prior technical training as GeneXpert MTB/RIF® super-users from Cepheid®. The team used this knowledge to develop a concept for preparing and producing GeneXpert MTB/RIF PT panels. The GeneXpert MTB/RIF assay cartridges and machines in the Uganda laboratory network were obtained from Cepheid® Inc with catalog number GXMTB/RIF-10. In 2014, the team used the known principle that the lowest detection level of GeneXpert MTB/RIF® technology was 100–1000 bacilli/ml in sputum [20] to conclude the required bacterial load concentration in the liquid panels. With this knowledge, McFarland 1.0 standard suspensions were prepared from Löwenstein Jensen (LJ) positive cultures and diluted

to 5.0 x$10^4$ CFU/ml that was used to prepare PT panels (liquid PT panels) after inactivation using the autoclaving technique.

## Staff training, SOP development, and ISO/IEC 17043 standard alignment

Staff who participated in the concept development and experiment in 2014 validated the procedure at the Uganda NTRL/SNRL in 2015 becoming pioneers and initiators of GeneXpert PT preparation. These were responsible for training, assessing the competence of additional personnel, and developing standard operating procedures. The initial SOP for GeneXpert MTB/RIF PT panel preparation was developed in 2014 and later revised in 2016 and 2017 to conform with ISO 17043 accreditation requirements to ensure quality panels were continually produced. The revision involved a break down of the original single SOP document to separate working documents. These included SOP for proficiency testing plan, instructions to participants, PT preparation and testing, data analysis, packaging, labeling, and distribution. The implementation of ISO/IEC 17043 further took shape with performance of internal audits twice a year to assess adherence to the set standards. The non-conformities identified from these audits were subjected to root cause analysis and subsequent corrective actions were addressed within given a period. Adherence to the clauses of the ISO/IEC 17043 standard lead to the assignment of a coordinator of the GeneXpert PT scheme and creating a PT team. Strict timelines for PT preparation procedures and other processes were drawn that streamlined practice and improved the QMS as regards PT preparation. This process later lead to an application process for accreditation to ISO/IEC 17043 of the South African national accreditation system (SANAS).

## PT panel production

Stock strains with known phenotypic and genotypic Polymerase Chain Reaction (PCR) sequencing results were used to produce the panels. A repository of these selected strains was kept at ultra-low freezer temperature (-80C). Negative strains were selected from *Mycobacterium fortuitum* which is a non-tubercle mycobacterium strain (NTM), internally prepared Phosphate Buffer solution (PBS), and sterile distilled water.

During the panel preparation phase, samples of three varying GeneXpert MTB RIF test results were incorporated. These included: (i) Negative strain; MTB Not Detected, (ii) Positive strain; MTB Detected: Rifampicin Sensitive and (iii) Positive strain; MTB Detected: Rifampicin Resistant.

After selection of these well-characterized Mycobacterium tuberculosis complex (MTBC) and Nontuberculous mycobacteria (NTM) strains, they were inoculated and grown on LJ culture media for 3–4 weeks to achieve the desired confluent growth. These biological samples used were not obtained through a medically prescribed test. They were got from an established bio-bank of the Institute of Tropical Medicine (ITM) WHO-Supranational Reference Laboratory Belgium (https://www.itg.be/E/administrator-of-the-worlds-largest-collection-of-tbc-strains) and each was accompanied with a conformity certificate as shown in "S1–S3 Texts" attached in the appendices.

The strains used were not specifically obtained for this study but also used for internal Quality controls during routine M.*tb* culture, phenotypic and genotypic drug susceptibility testing. Cultures were grown in quadruplicates and only those with colony growth (+3) according to Global Laboratory Initiative (GLI) guidelines were selected for use in producing strain dilutions [21, 22].

Strain suspensions of 1.0 Mcfarland standard were prepared from freshly grown cultures and heat-inactivated using an autoclave at 121pa for 30minutes to make the initial strain

stocks. A volume of 0.5ml of the inactivated suspension was inoculated on Mycobacteria Growth Indicator Tube (MGIT) and Lowenstein-Jensen (LJ) culture medium. The inoculum was monitored for 42 days and 8 weeks on MGIT and LJ respectively to prove the success of inactivation and ensure the safety of the personnel handling and testing the panel. Five pretest runs per sample stock were made to generate preliminary results which were to be compared with the validation results after proof of inactivation run.

On completion of proof of inactivation, aliquots of 0.5 mL of 0.5Mcfarland standard were added to 9.5 ml of sterile water to obtain a bacterial concentration of $1.0 \times 10^6$ bacilli/mL. Using the concentration of $1.0 \times 10^6$ bacilli/mL, an aliquot of 0.5 mL was added to 9.5 mL of sterile water to obtain a concentration of $5.0 \times 10^4$ bacilli/mL as the final dilution for the panels. This concentration when tested on GeneXpert MTB/RIF gives a medium-range quantification of M.tb per ml with a cycle threshold value between 16–22. Five aliquots of 1.0 mL of bacterial concentration for each strain were made into cryovials and validated on GeneXpert MTB/RIF assay to ascertain the quality and consistency with the intended results. All procedures above were conducted in a functional biosafety level III laboratory following developed SOPs and ISO/IEC 17043 standard.

Each panel was composed of four known liquid pretested and validated samples. The samples were completely de-identified from their original strain names to Q1, Q2, Q3, and Q4 labeling before validation and shipping to the participating laboratories. The Colony Forming Unit (CFU) count/ bacilli Load per sample was determined through panel pretest and validation using the GeneXpert MTB/RIF® assay. Although the amounts of detectable DNA varied from sample to sample, the panel composition was made from stocks that only yielded a cycle threshold (Ct) value of 16–22 (medium) during validation and pretesting.

Validation involved testing five-panel items from the stock of each sample that was chosen to be included in the panel. Inclusion of a sample in a panel was only after achieving 100% accuracy agreement among its validation test runs. Panels were freshly made for each round with uniqueness in composition i.e. the number of positive and negative samples, as well as the rifampicin sensitive and resistance, was distinctive for each new round.

Triple packaging was ensured by placing cryovial tubes into zip-lock bags that were wrapped in envelopes for sites within Uganda and transport boxes for those outside Uganda with adherence to the international air transportation association (IATA) regulations [23]. Each package was accompanied with the instructions to participants, confidentiality waiver, delivery acknowledgment, and result feedback form. The PT panels were sent as non-infectious shipments to the various laboratories. After shipment, a panel set was left at room temperature and tested after three weeks to check for stability and consistency. The three weeks were allocated on assumption that the latest panel sets would reach facilities after this period. This panel set was kept at ambient temperature and tested after three weeks from the date of dispatch. An IATA accredited courier company was sub-contracted for laboratories outside Uganda while to Ugandan laboratories, the ministry of health sample transport system was used. The above-described cascade of events was repeated for each round during the study.

A PT preparation and distribution schedule was established to streamline the scheme activities. Timelines were established for panel preparation, dispatch, and submission of feedback reports after closing dates. A period of 120 days (4 months) was allocated from panel preparation to dispatch of the panel sets to the testing sites, 42 days for laboratories to test and submit results back to NTRL, and 42 days to analyze data, prepare feedback, and round reports.

PT panels were sent out once in 2015 and 2016, twice in 2017 in February and August, and once in 2018. This schedule arrangement was put forward to give sufficient time for all the GeneXpert PT preparation and handling processes for each round conducted.

## Ethical statement

This was a mandated quality assurance initiative by National tuberculosis and leprosy control program to the NTRL to ensure quality services for all laboratories performing TB diagnostic work in the network. Approval was sought from the National Health Laboratory and Diagnostics services (NHLDS) review board and permission to publish study findings obtained from NTRL management under approval number NTRL/OR-2018001. It is worth noting that all enrolled sites were required to complete a Confidentiality Waiver and Participant Biosafety Compliance Letter Agreement before participation.

## Enrollment of participants in the PT scheme

This was dependent on the availability of a functional GeneXpert machine at the testing laboratories. An enrollment log was completed once annually with subsequent enrollment and recruitment of all laboratories housing functional machines within Uganda and outside Uganda. The enrollment was independent of the number of functional modules available for as long as a laboratory had at least a functional module. All laboratories without GeneXpert machines and those with all modules non-functional were excluded from enrollment in that particular round. All PT activities were centrally coordinated as shown in "**Fig 2**".

**PT pilot.**   The study started with a pilot that was conducted between October 2015 to May 2016. A sum of fifty-five testing laboratories was enrolled at this phase. These were selected from the different regions in Uganda. The selection was to give a comparison on how the different geographical dynamics and testing conditions would affect the performance of laboratories in GeneXpert PT.

**Round 1 (2016) to Round 4 (2018).**   After the pilot study, all the TB testing laboratories in Uganda were enrolled in the PT Scheme. The full enrollment was also a mandate of NTRL to provide quality assurance in the national laboratory network and the SRL-network. Enrollment forms sent out to participants captured their laboratory number and name, details of focal persons and location plus the status of QMS accreditation. The safety compliance details of all the laboratories enrolled were assessed during the process of GeneXpert machine installation to satisfy the minimum safety requirements to participate in the scheme.

For each new round of enrolment, the list of participants was updated. Both new and old participants were assessed for fitness to participant in the latest round. Testing was performed and results reported in the same format as in the pilot rounds. Scores were issued by NTRL in form of individual feedback reports for each site, detailing the breakdown of scores per tested sample. Individual performance feedback reports were sent through email and courier companies to the participants. Laboratories with unsatisfactory performances (defined as a panel score of less than 80%) were targeted for onsite supervision and followed up to do corrective actions.

## Results compilation and data analysis

Results obtained by participants from GeneXpert machines were manually entered into the reporting form that was sent along with the panels. The reporting form was then sent back through courier companies or electronically emailed to Uganda NTRL for analysis. Results were compiled and manually entered into controlled Microsoft excel version 2010 spreadsheets after submission at NTRL. The "IF command", "summation" and "Average" functions in excel were applied to award scores for each item tested by an individual participant and average testing turnaround time. This was also applied to calculate the final score. To evaluate trends of performance, graph charts were obtained using extracted summary data exported into a new excel spreadsheet. Laboratory performance was assigned an accuracy score

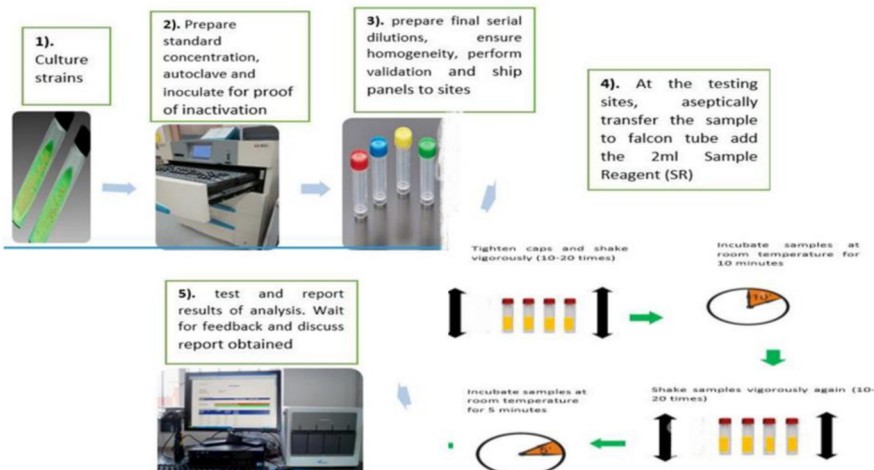

**Fig 2. Procedures during the GeneXpert MTB/RIF PT preparation and testing process.**

expressed as a percentage. Each of the four samples correctly tested accounted for 2 points, errors/invalids/no result machine readings were scored 1 point, and zero points were given for an incorrect result. The performance threshold for a satisfactory score was set at 7 out of 8 (7/8 points) which translated to a site correctly testing three samples and only scoring an error/invalid/no result for one of the four samples. The total points obtained were converted into a percentage that had to appear on the feedback reports shared with the participants.

To statistically understand the interaction of continuous participation of an individual laboratory and its performance in PT over rounds, we imported the data into SPSS version.24 and applied Ordinal logistic regression to determine the proportional odds. We hypothesized that PT performance would improve with continued participation. The association between PT performance (unsatisfactory or satisfactory) and the number of times a laboratory had participated was examined using the fixed- and mixed-effect proportional odds model [24]. GeneXpert MTB/RIF testing laboratories represented sampling clusters hence no random effects on the results obtained with conclusion done based on the fixed-effect model.

## Results and discussion

### Laboratory enrollment and participation

Five PT rounds were sent out between August 2015 and May 2018 as part of the GeneXpert MTB/RIF PT program of Uganda NTRL. A gradually increasing number of laboratories were enrolled from 55 sites in the pilot to 148 sites in Round 4(2018). In all the rounds, laboratories tested the panels using the GeneXpert MTB/RIF® assay.

**Table 1. Participation profile for all the rounds sent out.**

| Characteristics | Pilot study | Round 1(2016) | Round 2 (2017) | Round 3 (2017) | Round 4 (2018) |
|---|---|---|---|---|---|
| Laboratories enrolled | 55 | 110 | 121 | 127 | 148 |
| Return of results | | | | | |
| Total results returned n (%) | 51(93%) | 96 (87.3%) | 105(86.7%) | 104 (81.9%) | 134(90.5%) |
| No returns n (%) | 1(2%) | 10(9.1%) | 11(9.1%) | 20(15.7%) | 11(7.4%) |
| Non-functional machines | 3(5%) | 4(3.6%) | 5(4.2%) | 3(2.4%) | 3(2.1%) |

The set target for result submission was 80% of all participants enrolled per round. Overall, this was achieved with the lowest submission at 81.3% (104/127) in Round 3(2017).

In total, 18 sites reported being non-functional throughout the rounds with the highest number 5(4.2%) reported in Round 2(2017). The number of Laboratories not reporting after receiving PT panels varied from round to round, with Round 3(2017) having the highest no response rate of 20 (15.7%). The summary of participation is as shown in "**Table 1**" below.

## Performance of the participants

Laboratories achieving satisfactory and unsatisfactory scores were individually determined for each round. In the entire study, 490 results were generated with 93.5% (458/490) as satisfactory scores having the highest satisfactory performers 28.1% (129/459) in round 4 (2018). Round 1 (2016) saw the greatest proportion of unsatisfactory scores 37.5% (12/32) for all the rounds sent out. The majority of the participating laboratories failing at their first time of enrollment improved their performance in their subsequent participation while those initially scoring above 80% continued to match their original score. The lowest number of errors/invalids/no results was reported in Round 4 (2018) and highest in Round 1 (2016). Table 2 below summarizes the performance across all the rounds.

The median score improvement for a laboratory's first-time participation to its latest round of participation was 12 percentage for all laboratories, from 88% to 100%. On excluding laboratories that scored 100% at their first time of participation, the average score for the remaining facilities improved by 23%, from an average of 75% at the first time of participation to 98% in Round 4(2018).

An increasing sum of laboratories obtaining satisfactory scores increased with more times of participation as shown in **Table 3** below.

**Table 3** above shows laboratories that obtained satisfactory results during their proceeding times of participation. In the pilot study, 48/55 participants scored satisfactorily and this was taken as the baseline for all the laboratories that had been enrolled. In round 1(2016), 84/96 laboratories scored above 80% with 36 sites being their first time of participation and 48 their second time. In round 2 (2017), 98/105 laboratories scored satisfactorily with 51 these participating for the third time, 40 for the second time and 7 for the first time.

In round 3 (2017), of the 99 laboratories with satisfactory results 48 had participated consistently four times, 40 sites for three times, 6 two times and 5 laboratories at their first enrollment. Of the total 129/134 laboratories that had satisfactory results in round 4(2018), 50 were participating for the fifth time, 43 at their fourth time, 6 third time, 7 second time, and 23 were first-timers.

During these rounds of participation some sites didn't have satisfactory results at the first time of participation and later improved in the subsequent round increasing the number of satisfactory scores with each new round of enrollment which can also be noted in **Fig 3** below.

## Evaluation of the times of participation and unsatisfactory performance

A total of 32 unsatisfactory scores was generated from 24 laboratories throughout all the rounds. The main cause of unsatisfactory scoring was discordant result reporting at 68% (22/32) with the rest due to a laboratory reporting more than one error/invalids/no results. The error results reported were due to error codes 5006, 5007, 2008, and 2011. The percentage of sites achieving unsatisfactory scores generally decreased per round from 12.5% in round 1 (2016) to 3.7% in round 4 (2018) as shown in **Fig 3** above. The rate of scoring unsatisfactorily varied over times of participation as shown in **Table 4** below.

**Table 2. Performance of participating sites throughout the different rounds.**

| Performance | Pilot Study | Round1(2016) | Round 2 (2017) | Round 3 (2017) | Round 4 (2018) |
|---|---|---|---|---|---|
| Labs with Satisfactory score (≥80%), n(%) | 48(94%) | 84(87.5%) | 98(93.33) | 99(95.2%) | 129(96.3%) |
| Labs with Unsatisfactory score (≤80%), n(%) | 3(6%) | 12(12.5%) | 07(6.67%) | 5(4.8%) | 5(3.7%) |
| Errors/invalid/no result (n) | 00 | 14 | 06 | 11 | 02 |
| Discordant results | 03 | 06 | 05 | 10 | 07 |

Seventeen laboratories failed at their first time of participation, 9 at their second round of participation, and 5 at their third time from the pilot to round 4 (2018). For the fourth time of participation in round 3 (2017) and round 4(2018), no laboratory scored unsatisfactory results. At the fifth time of participation, only one laboratory failed among those that had been enrolled since the pilot study.

Of the three laboratories that scored unsatisfactory in the pilot study, one of these continued to score below 80% in round 1 (2016) and thereafter scored satisfactorily throughout the next rounds. In round 1 (2016), 12 laboratories scored unsatisfactorily with 9 of these participating for the first time and 3 for the second time. It is notable that 4/12 sites continued to score unsatisfactory in round 2 (2017) with one out of these four repeatedly failing in round 3 (2017) and thereafter all obtained satisfactory scores in round 4 (2018).

In round 2 (2017), 7 laboratories scored unsatisfactorily of which 2 were participating for the first time and 5 second time. All laboratories participating for the third time scored satisfactorily. In round 3 (2017), 5 laboratories scored unsatisfactorily with 2 of these being their first time of participation, one second time, and 2 for the third time. All laboratories that had failed in round 3 (2017) got satisfactory scores in round 4(2018).

In round 4(2018) that had 5 participants with unsatisfactory results, one of these was at their first time of enrollment, 3 were participating for their third consecutive time and one was at their fifth time of participation. The laboratory failing at its fifth time had been scoring satisfactorily in all the previous rounds it was enrolled.

In summary, only one laboratory consistently failed thrice, six sites twice and seventeen failed once. Of the seventeen laboratories that scored unsatisfactory once throughout their participation, ten of these scored at their first time of participation, three at their second consistent time of participation, three at their third participation time, and only one at its fifth participation round.

The odds of achieving scores of above 80% were estimated to be 2.51 [95% confidence interval (CI), 1.78 to 4.34] times greater with each new consistent round of participation. From this, we can say that a laboratory's performance in PT has higher chances of improving the more it participates. Our proportional odds regression model predicts that 91% of sites should be expected to achieve satisfactory scores (≥80%) after two rounds of prior experience with GeneXpert PT participation, and all should be expected to achieve 100% scores after four rounds of prior experience as also shown in **Fig 4** below.

**Table 3. Number of laboratories achieving satisfactory scores and their different participation times.**

| Time(s) of participation | Pilot study | Round 1 (2016) | Round 2 (2017) | Round 3 (2017) | Round 4 (2018) |
|---|---|---|---|---|---|
| First | 48 | 36 | 7 | 5 | 23 |
| Second | | 48 | 40 | 6 | 07 |
| Third | | | 51 | 40 | 6 |
| Fourth | | | | 48 | 43 |
| Fifth | | | | | 50 |

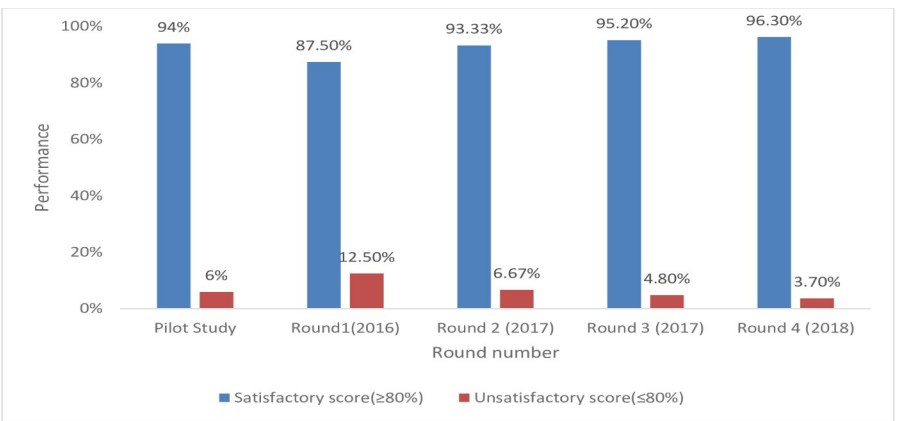

**Fig 3. Shows the performance of participants during the different round enrolled.**

## Discordant results

The highest recorded discordant result was the false-negative *M. tuberculosis* 35.4% (11/31) and the least being false rifampicin resistance at 19.4% (6/31) reported in all the rounds. There was no observed pattern between the number of errors/invalids/no results and the PT rounds.

From **Table 5**, 31 discordant results were recorded by 22 laboratories. One discordant result per laboratory was reported in rounds pilot study, round 1 (2016) and round 2 (2017) among the discordant performers. Round 3 (2017) registered two laboratories with 3 discordant results each with the rest being reported by one laboratory. Round 4 (2018) saw two laboratories reporting two discordant results each and the remaining three getting individually reported by three participants.

## Turnaround time (TAT)

From the 490 results from all the rounds, 78.8% (386/490) were received at NTRL within the set TAT of 42 days. The highest number of sites that submitted results within TAT was in round 4(2018) at 33.4% (129/386) just after round 3(2017) that registered the highest number of laboratories submitting results after established TAT 10.6% (11/104).

The average time to submit results by participants showed an improving trend from 30 days in the pilot to 18.5days in Round 4(2018) as shown in **Fig 5**. The study had its longest time taken to receive samples being 14 days with the highest contributors of this fate residing in the north-eastern part of Uganda. This can be accounted on for the long-distance these sites scale from NTRL. The time taken to receive samples in this study showed no correlation with obtaining discordant results or scoring unsatisfactorily.

## On and off-site Support Follow up (OSF) and corrective actions

Result reports were reviewed by competent personnel in the GeneXpert PT team before their dispatch to the respective participants. It is at this phase where laboratories that had failed

**Table 4. Laboratories scoring unsatisfactorily and their times of participation into the PT scheme.**

| Times of participation | Pilot study | Round 1 (2016) | Round 2 (2017) | Round 3 (2017) | Round 4 (2018) |
|---|---|---|---|---|---|
| First | 3 | 9 | 2 | 2 | 1 |
| Second | | 3 | 5 | 1 | 0 |
| Third | | | 0 | 2 | 3 |
| Fourth | | | | 0 | 0 |
| Fifth | | | | | 1 |

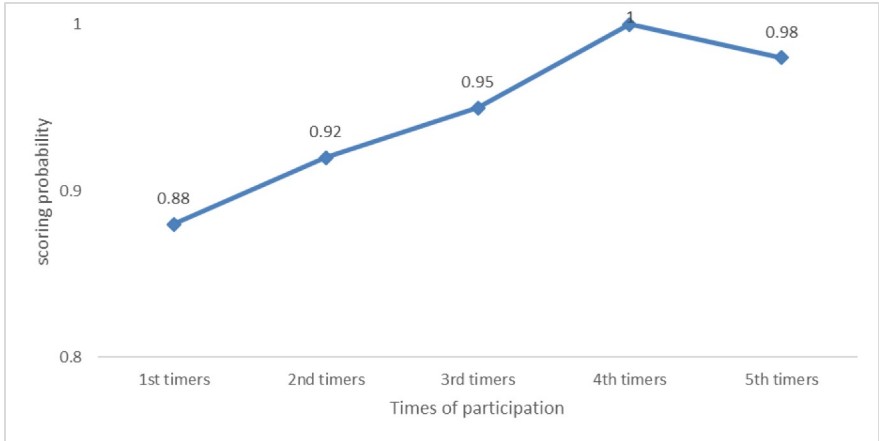

**Fig 4. Probability of laboratories scoring satisfactorily in GeneXpert PT over their continued rounds of participation.**

(scored unsatisfactorily) were assigned to individuals technical people at NTRL for follow-up. During annual supervision, NTRL personnel were able to follow up 23 laboratories onsite and support them in conducting root cause analysis and documenting corrective actions. The rest of the laboratories not visited due to logistic limitations were contacted and urged to submit their root cause analyses and corrective actions. Of the sites supervised, 17(90%) managed to score satisfactorily in their next round of participation signifying the positive impact of onsite follow-ups. Root cause analysis and corrective actions were not done for three (3) laboratories and these were among those not offered onsite follow-up. The mentioned three (3) above continued to score below 80% for the next round they participated in. The following issues as shown in **Table 6** were listed by 29 laboratories to be the causes for their failure.

Among the Issues identified during OSF and root cause analysis, clerical errors ranked highest for the unsatisfactory scores. This matter was cited among laboratories reporting false M.tb detection, false rifampicin sensitive, false rifampicin-resistant, and false-negative results. Submission of corrective actions from laboratories was challenging and a majority of sites had to be followed up over time to submit.

## ISO/IEC 17043 implementation

Documents and all records which included participants' results, preparation worksheets, SOPs, and personnel files generated during the rounds sent out, were used to apply for ISO 17043 accreditation. The application was successful and the scheme was accredited in 2018 SANAS. The adherence to this standard streamlined the follow of activities and promoted the quality of the PT program.

**Table 5. Discordant results reported from the different rounds.**

| discordant results | Pilot study | Round 1(2016) | Round 2 (2017) | Round 3 (2017) | Round 4 (2018) | Total |
|---|---|---|---|---|---|---|
| Laboratories reporting discordant results | 3 | 6 | 5 | 3 | 5 | 22 |
| false *M.tb* negative | 1 | 2 | 1 | 3 | 4 | 11 |
| false *M.tb* detected | 0 | 2 | 2 | 2 | 1 | 7 |
| false Rifampicin sensitive | 0 | 0 | 1 | 4 | 2 | 7 |
| false Rifampicin resistant | 2 | 2 | 1 | 1 | 0 | 6 |

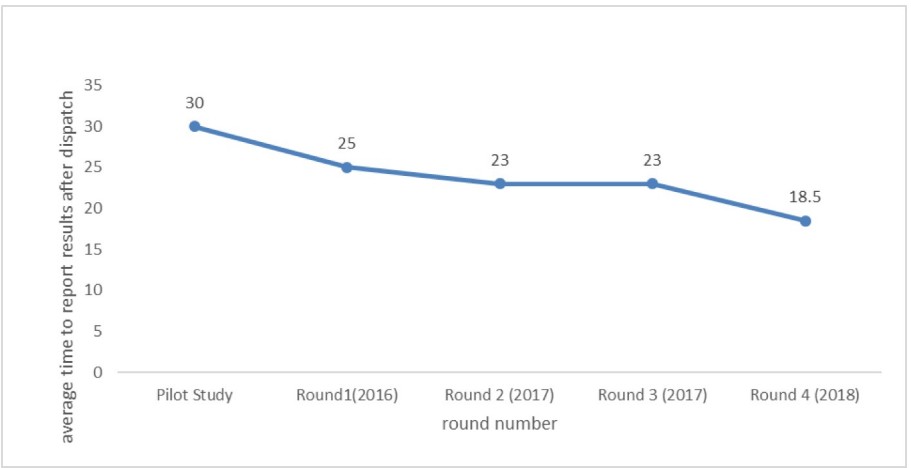

**Fig 5. Shows average TAT taken for sites to report results from the dispatch date during the different rounds.**

## Discussion

The introduction of GeneXpert MTB/RIF® technologies intensified case finding for TB. This is as a result of its improved sensitivity and turnaround time mainly in areas that have for long relied on sputum smear microscopy as the primary TB diagnostic [25]. By 2018 Uganda had a total of over 148 GeneXpert sites with 100% enrollment into the NTRL PT scheme. This number was supplemented by facilities outside Uganda that happen to be supported as a WHO Supra-national Reference Laboratory. The increase in the number of PT enrolled sites every year was because of the installation of more GeneXpert machines in Uganda and more laboratories outside Uganda applying for ISO 15189 accreditation that required them to be participating in a PT scheme.

The satisfactory performance of sites can be attributed to the competency at testing sites, the positive role of the QMS being implemented at several sites [26], proper functionality of the machines that were regularly serviced and well maintained daily.

The causes of unsatisfactory scores are several and these vary from one laboratory to another. These were related to human error, equipment maintenance, staff training and competency, environment and procedures being used. To a less extent, a false result could be attributed to the PT preparation process regardless of the rigorous quality control procedures undertaken as highlighted by other PT studies [27] and not the case in this one. Therefore, to

**Table 6. Frequencies of issues identified during root cause analysis for laboratories with unsatisfactory scores.**

| Issue identified | Frequency |
| --- | --- |
| Clerical reporting | 24.1% (7/29) |
| Poor expert machine maintenance | 10.3% (3/29) |
| Overdue instrument calibration | 13.8% (4/29) |
| Power blackout and lack of power backup | 20.7% (6/29) |
| Lack of air conditioners | 10.3% (3/29) |
| Lack of staff training and orientation of new staff | 6.9% (2/29) |
| Sample pouring | 3.4% (1/29) |
| GeneXpert cartridge stock out | 3.4% (1/29) |
| Lack of a printer | 3.4% (1/29) |
| Switching of samples | 3.4% (1/29) |

unearth the possible cause of performance failure in a PT, one has to holistically do unbiased root cause analysis and identify the only possible cause if it is not so obvious.

The commonest errors reported in this study were 5006, 5007, 2008, and 2011 that are due to Probe Check failures caused by high sample viscosity and/or low volumes and communication loss between the GeneXpert machine and the computer. These were followed by invalid results which occur due to Internal Control failure instigated by several factors such as improper kit and sample storage and inadequate sample volume [28].

The majority discordant result (false negative) *M. tuberculosis complex* results can be caused by PT samples having small amounts of detectable *M. tuberculosis* DNA. This is resultant from sample degradation caused by the addition of insufficient PT sample volume to test cartridges or other procedural errors such as improper sample storage and cross-contamination. Other reasons can be clerical reporting on the result form and switching of samples during testing. These same aspects in this study were also reported in the article "Development, evaluation, and implementation of a new rapid molecular diagnostic for tuberculosis and rifampicin resistance" [29]. This highlights the picture of the possible errors in the analytical cascade for TB clinical specimen vis-à-vi GeneXpert MTB/RIF® assay utilization. The major cause of all false results was clerical reporting caused by manual transcription of results to the provided reporting template. To overcome this challenge, PT participants should employ and strengthen their result review process before reporting to the PT provider. This system should directly be applied to the routine clinical samples to mitigate transcription errors. The performance of laboratories within and outside Uganda had no significant difference and scores were seen not to vary at all in regards to the location of the laboratory.

The Turnaround time (TAT) in PT can be reasonably related to the TAT taken to report patient results. Timely feedback in EQA is crucial to the program's effectiveness. Delayed result submission not only obstructs the timely implementation of essential corrective actions but also prolongs poor testing accuracy, thereby influencing the quality of patient care and EQA relevance in the end. The mechanism of PT delivery and result sending in this study was found to highly affect the time taken to test and report results.

The main courier company used at the time of the study was not highly effective in reaching every laboratory as quickly as possible causing delays in shipment and testing of samples. The great improvement in round 1 (2018) TAT was due to improved adherence to instructions sent to the participants and increased use of emails to submit results compared to the earlier rounds.

The low response rate and many sites reporting after set TAT in round 3(2017) were as a result of NTRL transitioning from one courier company to another that the participants had not yet fully got accustomed to. The delays were seen more in the Ugandan laboratory network where PT materials never got to reach their intended destinations within ten working days that were always allocated for dispatch and receipt of PT materials. This, therefore, implies the need for a robust but flexible transport system that reaches up to final laboratories directly from the PT provider. The other solution is enrolling out an online result submission system that bypasses the need to submit results through courier. In this study, the emailing system was employed more in the last round to help in receiving feedback results in addition to the courier. This contributed to the improvement in response rate.

Activities performed during on-site supervision ranged from refresher pieces of training and competence assessment of personnel, review of laboratory registers, EQA files, GeneXpert machine maintenance logs, and troubleshooting for errors/invalids/no results generated. The Authorized GeneXpert Servicing contractor by Cepheid in Uganda handled all the responses and matters that were beyond the technical knowledge of NTRL personnel such as module replacement, calibration, and software upgrades. All these actions need to be routinely done to

ensure consistency of quality results in a laboratory network. The majority of the problems listed under root cause analysis were instantly fixed. Suggestion and actions taken included **the** development of temperature monitoring and Xpert machine maintenance logs for sites without them. Laboratory managers were assigned to compile maintenance logs monthly, schedule timely Xpert instrument calibration every year, and improve supply chain management for the supply of cartridges to fix the challenges raised. Actions that required more funding included the installation of power backups such as solar panels and UPS, replacement of failed modules, installation of air conditioners, and procurement of result printers. These were forwarded to the National TB and Leprosy Program(NTLP) to be fixed. Fixing of actions that required many funds was never achieved in short times and affected the performance and routine effectiveness of the laboratories.

To further improve the GeneXpert PT program, some technical personnel were trained by the Centers for Disease Control (CDC), Atlanta in July 2017 to enroll Dry Tube Specimen (DTS) that has better performance characteristics as compared to the liquid panel that was provided in these rounds.

The establishment of a quality management system based on ISO 15189 for medical Laboratories and the 17043 PT program and having their accreditation is fundamental in sustaining a PT program and having a quality output.

## Conclusion

The results of this implementation study reveal the gaps and roadmap needed to establish a GeneXpert PT scheme in countries with similar settings and bottlenecks. The results clearly show the positive impact of continued PT participation and how it improves testing accuracy and consistency at the TB testing laboratories. The study also shows that there is a great value of following up sites with unsatisfactory performance as this enables an efficient fix and gives a long-term solution to occurrences at the laboratories with issues regarding their quality management system, with improvement in their performance. The findings highlight the feasibility of implementing a GeneXpert PT Scheme, and the need to reduce the turnaround time for all the steps involved in the same. The lessons learned may be helpful for other countries to replicate successfully in their settings.

## Supporting information

**S1 Text. Conformity certificate of biological samples used to prepare PT panels.**
(PDF)

**S2 Text. Conformity certificate of biological samples used to prepare PT panels.**
(PDF)

**S3 Text. Conformity certificate of biological samples used to prepare PT panels.**
(PDF)

**S1 File.**
(PDF)

**S2 File.**
(PDF)

**S3 File.**
(PDF)

**S4 File.**
(PDF)

**S5 File.**
(PDF)

## Acknowledgments

We thank the study participants for their time, interest, and willingness to and the staff of the National TB Reference Laboratory of Uganda who we applaud for the technical and administrative work. We acknowledge the training and advisory support from and Regional Global fund through the East, Central, and Southern Africa (ECSA) Health Community Project.

## Author Contributions

**Conceptualization:** Joel Kabugo, Joanita Namutebi, George William Kasule, Abdunoor Nyombi, Moses L. Joloba.

**Data curation:** Joel Kabugo, Joanita Namutebi, Dennis Mujuni, Andrew Nsawotebba, Edgar Kigozi.

**Formal analysis:** Joanita Namutebi, George William Kasule.

**Funding acquisition:** Kenneth Musisi, Moses L. Joloba.

**Investigation:** Joel Kabugo, Joanita Namutebi, Dennis Mujuni, Andrew Nsawotebba, George William Kasule.

**Methodology:** Joel Kabugo, Joanita Namutebi, Dennis Mujuni, Andrew Nsawotebba, Abdunoor Nyombi, Fredrick Kangave.

**Project administration:** Joel Kabugo, Joanita Namutebi, Kenneth Musisi, Pius Lutaaya, Fredrick Kangave.

**Resources:** George William Kasule, Kenneth Musisi, Fredrick Kangave, Moses L. Joloba.

**Supervision:** Joel Kabugo, Joanita Namutebi, Dennis Mujuni, Andrew Nsawotebba, Pius Lutaaya.

**Validation:** Joel Kabugo, Joanita Namutebi, Dennis Mujuni, Andrew Nsawotebba, Pius Lutaaya.

**Visualization:** Joel Kabugo, Edgar Kigozi.

**Writing – original draft:** Joel Kabugo.

**Writing – review & editing:** Joel Kabugo, Joanita Namutebi, Dennis Mujuni, Andrew Nsawotebba, George William Kasule, Kenneth Musisi, Edgar Kigozi, Abdunoor Nyombi, Moses L. Joloba.

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
