## [Decision Letter · Decision Letter 0]

1 Sep 2020

PONE-D-20-20339

Implementation of GeneXpert MTB/Rif proficiency testing program: A Case of the Uganda national tuberculosis reference laboratory/Supranational reference laboratory

PLOS ONE

Dear Kabugo,

Thank you for submitting your manuscript to PLOS ONE. After careful consideration, we feel that it has merit but does not fully meet PLOS ONE’s publication criteria as it currently stands. Therefore, we invite you to submit a revised version of the manuscript that addresses the points raised during the review process. In addition to reviwers comments, it is noted that the text has similarity to atleast one other study PMID : 30567748. So, the text must also be revised for oroginality. 

We look forward to receiving your revised manuscript.

Kind regards,

Pradeep Kumar, Ph.D.

Academic Editor

PLOS ONE

Journal Requirements:

2. Please upload a copy of Figures 1-5, to which you refer in your text. If the figures are no longer to be included as part of the submission please remove all references to them within the text.

Reviewers' comments:

Reviewer's Responses to Questions

**Comments to the Author**

1. Is the manuscript technically sound, and do the data support the conclusions?

Reviewer #1: Yes

Reviewer #2: Partly

2. Has the statistical analysis been performed appropriately and rigorously? 

Reviewer #1: I Don't Know

Reviewer #2: Yes

3. Have the authors made all data underlying the findings in their manuscript fully available?

Reviewer #1: Yes

Reviewer #2: Yes

4. Is the manuscript presented in an intelligible fashion and written in standard English?

Reviewer #1: No

Reviewer #2: Yes

5. Review Comments to the Author

Reviewer #1: • What is the reason behind author’s choice for using autoclaving as an inactivation method when SR reagent is a proven method of inactivation for Mtb? Did the authors conduct any studies to verify that autoclaving does not impact the GeneXpert results?

• Further, authors note that samples are stored at 2-8C for 42 days until results from MGIT are verified for inactivation. Does this long storage of samples have any impact on the results? Were any experiments conducted?

• It is noted in the supplementary section that samples are shipped at ambient temperature. Were overnight shipment methods used? What was the longest it took for samples to arrive at a designated laboratory? Was there any correlation with long transit times and discordant/invalid results? In other words, did long transit times lead to sample degradation and hence invalid results?

• Authors are requested to provide an original reference for lines 110-111 for the detection limit of GeneXpert MTB/RIF test.

• Under supporting information, authors listed figures 1 to 5 as S1 Fig 1, S2 Fig 2 etc., Please clarify which figures belong in the main manuscript and supplementary sections respectively.

• There is no information about the process behind obtaining ISO accreditation. Some detail about what changes were made to the SOPS to align with ISO’s requirements and how this aided in study’s success supporting lines 438-439 would be helpful.

• Authors note that the major cause of false results is clerical errors. Did the authors consider using automated GeneXpert test reports instead of manual reporting?

• Were the panel of strains freshly prepared during each around? Please clarify in methods section.

• How are the results reported when a test reports no MTB detected but RIF sensitivity/resistance is detected?

Minor comments:

• Authors are requested to expand the acronyms the first time they appear in the manuscript.

For example, PT in line 25, EQA in line 61, QMS in line 85 etc.,

• Similarly, authors use proficiency testing and PT interchangeably throughout the manuscript. For consistency authors should either stick to the acronym or the expanded version not both.

• Authors are advised to thoroughly revise the manuscript for grammatical errors and sentence formation for clarity in several areas.

• In lines 351-353, machine calibration is mentioned twice as one of the issues.

• Please rephrase lines 27-28 of abstract to “conducted between years 2015 to 2018” for clarity

Reviewer #2: Overall

This paper reports data from a cross-sectional study assessing the testing accuracy of the GeneXpert MTB/RIF assay in laboratories linked to the Uganda National Tuberculosis Reference Laboratory and assessing whether a program of GeneXpert proficiency testing in Uganda led to improved testing accuracy among participating laboratories. As the authors well describe in the introduction, national TB programs in resource-limited countries, especially in sub-Saharan Africa, have not generally established proficiency testing for GeneXpert MTB/RIF. This is a novel and interesting piece of work because it shows that setting up a persistent country-wide proficiency testing program for GeneXpert can lead to increased testing accuracy through root cause analysis and assistance and advice from a reference laboratory. However, a number of important inconsistencies and errors in the results section need to be corrected before the manuscript is suitable for publication, and I have concerns about the 80% threshold used to judge a laboratory as satisfactory. The manuscript would also benefit from a more in-depth discussion of the headline result that average laboratory scores increased the more rounds of the testing program they participating in. The article is intelligible but would nevertheless benefit from independent editorial assistance to improve the style and occasional grammatical errors. Below are my numbered comments and recommendations:

Introduction

1. Line 61: EQA should be defined as it is its first use.

2. Overall, the introduction explains the background to the study well, including the Uganda-specific situation and gives a good definition of proficiency testing.

Methods

I am unable to comment on the methods of the PT panel preparation as this is outside my area of expertise.

3. Line 242-4: I don’t understand why the scoring threshold in set at 80%, or even expressed as a percentage at all, because it gives the impression of greater accuracy than is truly available from your testing system, in which each laboratory tests 4 samples. 6 out of 8 points gives 75% (unsatisfactory) and 7 out of 8 gives 87.5% (satisfactory). So to be judged satisfactory, laboratories have to correctly test all 4 samples (100%), or correctly test 3 samples and have one error/invalid/no result (87.5%), but will automatically be judged unsatisfactory if they give one of more incorrect results. This system is fine, but it would be better would be to express the threshold as 7 out of 8 points for clarity.

4. Currently the names of the study rounds are very confusing and difficult to keep track of, with round 2 followed by round 1 followed by round 2 followed by round 1. Much better would be to make the numbering sequential, along the lines of: Pilot study, Round 1 (2016), Round 2 (2017), Round 3 (2017), Round 4 (2018). Or alternatively: Pilot study, 2016, 2017a, 2017b, 2018.

5. I have no ethical concerns, or concerns about the statistical methods.

Results

6. There are significant discrepancies between the numbers in Table 1 and Table 2. For the pilot study, Table 1 says that 48 results were returned but Table 2 has 48+3=51 results. Round 2 2016 and Round 1 2017 have consistent numbers between the two tables, but Round 2 2017 has 104 results returned in Table 1 but 100+5= 105 results in Table 2 and Round 1 2018 has 134 results returned in Table 1 but 115+4=119 results in Table 2. So 3 out of 5 are inconsistent.

7. Additionally, Line 288 says: “There were a total of 19 unsatisfactory results form 15 laboratories”. However, according to Table 2, there were 3+12+7+5+4=31 unsatisfactory scores. And then line 347-8 says: “3 out of the total 19 sites with unsatisfactory scores…”

8. Lines 308-310: These results are not in fact shown in Table 2 and Figure 3. Table 2 shows 29 discordant results, not 23. Presumably, those additional 6 discordant results came from laboratories which had more than one discordant result. Table 2 would be more useful if updated to show the number of laboratories with at least one discordant result in each round, rather than the total number of discordant results, possibly with the breakdown of false positives/negatives/resistant/sensitive below.

9. Lines 305-7: These lines are not results and should be moved to the discussion section.

10. Line 337-9: These lines are not results and should be moved to the discussion section.

11. Lines 275-6, 292-4, 298-300 and 310-11: These points are very interesting, and the odds ratio of 2.5 is one of the major results you report in the abstract, so I feel they deserve a graph or table of their own, showing the average scores (or number of satisfactory and unsatisfactory results) of laboratories in the first round they engaged in, and the second round, third round and fourth round for those which engaged in multiple rounds. One purpose of the study was to see whether continual participation in the proficiency testing program would improve the quality of TB testing results, which it appears to, and this graph or table would present this data in the clearest way.

12. Line 350-360: This list of issues could be made more useful if the frequency with which each issue arose was displayed.

13. It would be useful to see results broken down by whether the laboratory was inside or outside Uganda. I can see potential for the impact of the proficiency testing to be different for laboratories outside Uganda. Or if there was no significant difference, add a comment to say there was no difference.

6. PLOS authors have the option to publish the peer review history of their article (what does this mean?). If published, this will include your full peer review and any attached files.

Reviewer #1: No

Reviewer #2: No

---

## [Author Response · Author response to Decision Letter 0]

13 Dec 2020

I extend my appreciation to the precious time and dedication given to my manuscript. The reviews and comments raised have been taken positively and as authors of this manuscript, we would like to respond to them as below;

Review Comment: It is noted that the text has similarity to atleast one other study PMID : 30567748. So, the text must also be revised for originality

Response: The manuscript has been exclusively revised to exhibit originality of research and bring out the innovation of using liquid panels for GeneXpert proficiency Testing panels with clear distinction from the research work presented in study PMID : 30567748

Response to Reviewer 1

1. What is the reason behind author’s choice for using autoclaving as an inactivation method when SR reagent is a proven method of inactivation for Mtb? Did the authors conduct any studies to verify that autoclaving does not impact the GeneXpert results?

Yes, we conducted a preliminary experiment by inactivating eight samples using either method and inoculating on both LJ culture and MGIT culture after inactivation. This was done at the initiation of this study. The preliminary results obtained from inactivation of samples for GeneXpert proficiency testing, showed that the method of using sample reagent (SR) buffer to inactivate samples yielded less effectiveness than the autoclaving method. Proof of inactivation results gave two sample strains inactivated using sample reagent (SR) showing growth on both MGIT method and LJ culture method at the 5th week of culturing while all those inactivated using autoclaving method showed no growth. 

2. Further, authors note that samples are stored at 2-8C for 42 days until results from MGIT are verified for inactivation. Does this long storage of samples have any impact on the results? Were any experiments conducted?

Yes, experiments were done to ensure stability and consistency. For every round prepared, pretest of five runs on inactivated sample was done before storage at 2-8°C and after the forty two days a validation test was done to prove consistency of M.tb DNA in the samples as highlighted on line 162-164 and 169-171. The results showed no difference on the expected results.

3. It is noted in the supplementary section that samples are shipped at ambient temperature. Were overnight shipment methods used? What was the longest it took for samples to arrive at a designated laboratory? Was there any correlation with long transit times and discordant/invalid results? In other words, did long transit times lead to sample degradation and hence invalid results?

In this study no overnight shipment methods were used. The longest it took samples to reach final destination was fourteen days. The time taken to receive samples in this study showed no correlation with obtaining discordant results as highlighted in line 371-374. To ascertain whether our samples were still stable and consistent even after shipment were performed a stability test on a set of panel for each round. This panel was kept at ambient temperature and tested after three weeks from date of dispatch as shown in line 193-198.

4. Authors are requested to provide an original reference for lines 110-111 for the detection limit of GeneXpert MTB/RIF test

This has been added on line 111 (citation 21; Study done by Danica Helb et, al in 2010

5. Under supporting information, authors listed figures 1 to 5 as S1 Fig 1, S2 Fig 2 etc., Please clarify which figures belong in the main manuscript and supplementary sections respectively.

Figures 1-5 all belong to the main manuscript and have been added.

6. There is no information about the process behind obtaining ISO accreditation. Some detail about what changes were made to the SOPS to align with ISO’s requirements and how this aided in study’s success supporting lines 438-439 would be helpful

Line 120-134 have been added in methods to detail the process behind ISO 17043 accreditation. Line 402-407 have been added also to show the impact of aligning to ISO 17043.

7. Authors note that the major cause of false results is clerical errors. Did the authors consider using automated GeneXpert test reports instead of manual reporting?

During this study, we noticed the need to enroll an electronic system but we were unable to enroll it out during the study period. The system has developed in 2020 and piloted in Uganda but not yet put to full use.

8. Were the panel of strains freshly prepared during each around? Please clarify in methods section.

Panels were freshly prepared for each round as highlighted on line 185-187.

9. How are the results reported when a test reports no MTB detected but RIF sensitivity/resistance is detected?

The score was zero (0) for a site that reported “no MTB detected but RIF sensitivity/resistance is detected”. This being that rifampicin resistance could not be expected without MTB detection.

10. Minor comments

All acronyms have been expanded the first time they appear in the manuscript. For consistency acronym PT has been used in the entire manuscript. The entire manuscript has been revised to address grammatical errors. 

Response to Reviewer 2

1) EQA should be defined as it is its first use.

This has been addressed.

2) The scoring system 

For the scoring system, we have added the fraction score, e.g. 6 out of 8 points (line 262-266) in the methodology write up as suggested by the reviewer to clarify on the passing threshold. The adoption of a percentage score was to align the participant reports to the requirements of ISO 17043 clauses on data analysis that required percentage-scoring system.

3) Names of the study rounds

The round number have been renamed as; Pilot study retained its naming, round 2 2016 to round 1 (2016), round 1 2017 to round 2 (2017), round 2 2017 to round 3 (2017), and round 1 2018 to round 4 (2018).

4) Result discrepancy 

The results have been thoroughly re-analyzed from the raw data to address the discrepancy in the earlier submission as noticed by the reviewer. 

6a) The results in table 1 for pilot study were corrected to, 

parameter Before review After review

Total results returned 48 51

No returns 4 1

Non-functional sites 3 3

In table 2, the results remained as before for the pilot study as follows 

parameter Before review After review

Satisfactory score 48 48

Unsatisfactory score 3 3

Non-functional sites 3 3

In table 1, the results remained as before review for round 3 (2017)

The results in table 2 for round 3 (2017) were corrected as below, 

parameter Before review After review

Satisfactory score 100 99

Unsatisfactory score 5 5

In table 1, the results remained as before review for round 4 (2018)

The results in table 2 for round 3 2017 were corrected as below, 

parameter Before review After review

Satisfactory score 115(97.5%) 128(95.5%)

Unsatisfactory score 4(2.5%) 6(4.5%)

Line 297 (formerly line 288), number of times with sites scoring unsatisfactory was adjusted to 32 to match results in table 2 after data re-analysis.

Table 3 and Table 4 were added to show the number of laboratories scoring satisfactory and those with unsatisfactory respectively with their continuous times of participation. 

Figure 4 graph has been added after re-analysis to show probability of scoring satisfactorily given the number of times a laboratory consistently participates

Table 5 was added showing to breakdown the different discordant results (false positives/negatives/resistant/sensitive) 

5) List of issues raised during on and off site follow up

Table 6 has been added to show the frequency of issues raised during on and off site follow up

6) Performance of a laboratory inside or outside Uganda

The performance of laboratories within and outside Uganda had no significant difference and scores were seen not to vary at all in regards to the location of the laboratory as highlighted in line 446-448.

I would like again to appreciate the time awarded for this manuscript. 

Thanks PlosOne 

Yours faithfully

Kabugo Joel; 

Author Implementation of GeneXpert MTB/Rif proficiency testing program: A Case of the Uganda national tuberculosis reference laboratory/Supranational reference laboratory

---

## [Decision Letter · Decision Letter 1]

20 Jan 2021

PONE-D-20-20339R1

Implementation of GeneXpert MTB/Rif proficiency testing program: A Case of the Uganda national tuberculosis reference laboratory/Supranational reference laboratory

PLOS ONE

Dear Dr. Kabugo,

Thank you for submitting your manuscript to PLOS ONE. After careful consideration, we feel that it has merit but does not fully meet PLOS ONE’s publication criteria as it currently stands. Therefore, we invite you to submit a revised version of the manuscript that addresses the points raised during the review process.

Please addresss reviewers comments and I would like to draw attention to a thorough review of langauge.

We look forward to receiving your revised manuscript.

Kind regards,

Pradeep Kumar, Ph.D.

Academic Editor

PLOS ONE

Reviewers' comments:

Reviewer's Responses to Questions

**Comments to the Author**

1. If the authors have adequately addressed your comments raised in a previous round of review and you feel that this manuscript is now acceptable for publication, you may indicate that here to bypass the “Comments to the Author” section, enter your conflict of interest statement in the “Confidential to Editor” section, and submit your "Accept" recommendation.

Reviewer #1: All comments have been addressed

Reviewer #2: (No Response)

2. Is the manuscript technically sound, and do the data support the conclusions?

Reviewer #1: Yes

Reviewer #2: Yes

3. Has the statistical analysis been performed appropriately and rigorously? 

Reviewer #1: I Don't Know

Reviewer #2: Yes

4. Have the authors made all data underlying the findings in their manuscript fully available?

Reviewer #1: Yes

Reviewer #2: Yes

5. Is the manuscript presented in an intelligible fashion and written in standard English?

Reviewer #1: No

Reviewer #2: Yes

6. Review Comments to the Author

Reviewer #1: In this revised version of manuscript titled “Implementation of GeneXpert MTB/Rif proficiency testing program: A Case of the Uganda national tuberculosis reference laboratory/Supranational reference laboratory” authors addressed most of the comments made by the reviewers. Important corrections were made in the results section and additional tables were included as per the suggestions of one of the reviewers. However, there are few additional corrections and discrepancies noted in the results section that need to be addressed before the manuscript is acceptable for publication. Further, the manuscript requires thorough revision to address typographical, grammatical errors and paraphrasing in certain sections for clarity. Following are my comments:

• In table 2, the percentage for round 2 should be corrected to 93.33 instead of 94.1

• Numbers in tables 3 and 4 do not make sense. For example, in table 3, how did the labs participating for the third time increase to 51 in round 2. Also, numbers 50 and 43 in round 4 also do not add up.

• Similarly, in table 4 how did the laboratory scoring unsatisfactorily for the 5th time appear while there are zero instances of 4th time. The results in the table also does not match the text in lines 335-338. Authors are requested to carefully revise and address.

• Lines 168-172: Authors performed a 20-fold dilution from 1.0 x 10^6 bacilli/mL. This will achieve a concentration of 5 x 10^4 bacilli/mL and not 1.0 x 10^4 bacilli/mL. Authors are requested to clarify. What is the final concentration tested on geneXpert?

Minor comments:

• Lines 51-54 and 210-211 may be rephrased for clarity.

• Please explain the relevance of lines 153-155

• Table 5 needs formatting adjustments for the headers to match the columns. Also, for consistency, authors may follow similar table formatting throughout the manuscript.

• Figure 1 has a typographical error in steps 4 and 5 (contious). The word pursued in misspelled in line 34 of abstract.

• Capitalization of words in the middle of a sentence is noted in several areas of the manuscript. For example – line 27 (Panels); line 29 (Laboratories); line 64 (Support Supervision). Authors are suggested to revise such areas.

Reviewer #2: The manuscript is much improved from the previous version. Figure 4, Table 5 and Table 6 are all good additions to the manuscript. However, there are still some errors and inconsistencies in the tables. My numbered points are below:

1.

Table 2:

Discordant results read: 3, 6, 5, 10, 7

Table 5:

Discordant results read: 3, 6, 5, 3, 5

2.

Table 5: Column values for false negative M. tb add up to 11 (1+2+1+3+4) but total says 10

3.

Table 5: Round 4 (2017) should read Round 4 (2018)

4.

Line 368-9: “The lowest number of errors/invalids/no results was reported in Round

369 4 (2018) and highest in Round 1 (2016) with errors reported as shown in Table 5”

This is misleading because from Table 5 one would say that the lowest number was in the pilot study or Round 3 (2017) and the highest was in Round 1 (2016). I presume you’re talking about the errors/invalids/no results as shown in Table 1. This comment should therefore be moved to Table 1.

5.

Line 64: First use of PT in the text (as opposed to the introduction) so should be “Proficiency Testing (PT)”

6.

Line 97: “Mycobacteria. Tuberculosis complex” should be “Mycobacterium tuberculosis complex”

7.

Line 154: is this supposed to have an error message?

7. PLOS authors have the option to publish the peer review history of their article (what does this mean?). If published, this will include your full peer review and any attached files.

Reviewer #1: No

Reviewer #2: No

---

## [Author Response · Author response to Decision Letter 1]

10 Feb 2021

Response to Reviewer 1

1. In table 2, the percentage for round 2 should be corrected to 93.33 instead of 94.1 and unsatisfactorily adjusted to 6.67%. This has been done and also fig 3 adjusted

2. Tables 4 and 5 have been given further elaborations on the increasing numbers of participating scoring satisfactory scores as we moved through the rounds from line 323-327 and 352-372.

3. The results in the table also does not match the text in lines 335-338. Authors are requested to carefully revise and address. 

More revisions has been made and explanation for this made from line 352-372. Information in the table has been aligned to match with that in the paragraphs (352-362) formerly 335-338.

4. Lines 168-172: Authors performed a 20-fold dilution from 1.0 x 10^6 bacilli/mL. This will achieve a concentration of 5 x 10^4 bacilli/mL and not 1.0 

This has been amended to a concentration of 5.0 x 104 bacilli/mL as the final dilution for the panels since this is the actual dilution made. The target bacterial concentration of the panels was between 1.0 x 104 and 7.5 x 104 bacilli/mL.

5. Line 51-54 rephrased starting from line 51-58. 

The whole manuscript has been paraphrased

6. Line 153-155 now line 157-159, now line 158-60

Was a clarification request from the Plos one Editorial team to “declare whether the strains used in the study were specifically for this study only or not”.

7. All tables have been reformatted and made consistent

8. Figure 1 has been rephrased at step 4 and 5 correcting typographical errors and capitalization in the middle of sentences. 

9. Word Pursued in Line 34 has been corrected.

10. Other Words with capitalization in the middle of the sentence have been corrected with a more thorough manuscript revision by coauthors 

Reviewer 2

1) In table 5, the figures 3, 6, 5, 3, 5 corresponds to the number of laboratories that reported the 3, 6, 5, 10, 7 discordant results in table 2 during the different rounds in the study. Elaboration statements have been added below table 5 to bring out this information.

2) The total of false negative M. tb in table 5 is adjusted to 11 corresponding to the figures when added up

3) Renaming of rounds in table 5 has been made.

4) The statement “the lowest number was in the pilot study or Round 3 (2017) and the highest was in Round 1 (2016) has been moved to line 308-309.

5) PT written in full (Proficiency Testing) for its first appearance in text line 68 former line 64

6) Mycobacteria. Tuberculosis complex” corrected to “Mycobacterium tuberculosis complex” in line 101.

7) Line 153-155 now line 157-159, Was a clarification request from the Plos one Editorial team to “declare whether the strains used in the study were specifically for this study only or not”.

---

## [Decision Letter · Decision Letter 2]

16 Apr 2021

PONE-D-20-20339R2

Implementation of GeneXpert MTB/Rif proficiency testing program: A Case of the Uganda national tuberculosis reference laboratory/Supranational reference laboratory

PLOS ONE

Dear Dr. kabugo,

Thank you for submitting your manuscript to PLOS ONE. After careful consideration, we feel that it has merit but does not fully meet PLOS ONE’s publication criteria as it currently stands. Therefore, we invite you to submit a revised version of the manuscript that addresses the points raised during the review process.

We look forward to receiving your revised manuscript.

Kind regards,

Shampa Anupurba, MD

Academic Editor

PLOS ONE

Journal Requirements:

Additional Editor Comments (if provided):

Line 55- ' mycobacterium tuberculsosis ' to be corrected to 'Mycobacterium tuberculosis'

Line 168- ' mgit and lj' to be written as MGIT and LJ and also expanded.

Line 421- 'It is at this phase were...'- were may be replaced by 'where'

Reviewers' comments:

Reviewer's Responses to Questions

**Comments to the Author**

1. If the authors have adequately addressed your comments raised in a previous round of review and you feel that this manuscript is now acceptable for publication, you may indicate that here to bypass the “Comments to the Author” section, enter your conflict of interest statement in the “Confidential to Editor” section, and submit your "Accept" recommendation.

Reviewer #2: All comments have been addressed

2. Is the manuscript technically sound, and do the data support the conclusions?

Reviewer #2: (No Response)

3. Has the statistical analysis been performed appropriately and rigorously? 

Reviewer #2: (No Response)

4. Have the authors made all data underlying the findings in their manuscript fully available?

Reviewer #2: (No Response)

5. Is the manuscript presented in an intelligible fashion and written in standard English?

Reviewer #2: (No Response)

6. Review Comments to the Author

Reviewer #2: (No Response)

7. PLOS authors have the option to publish the peer review history of their article (what does this mean?). If published, this will include your full peer review and any attached files.

Reviewer #2: No

---

## [Author Response · Author response to Decision Letter 2]

24 Apr 2021

Content Review

Line 55- ' mycobacterium tuberculsosis ' has been corrected to 'Mycobacterium tuberculosis'

Line 165 and 166- ' mgit and lj' has been re-written as MGIT and LJ and also expanded.

Line 421- were has been replaced by 'where'

References review

Reference (2) has been re-referenced citing it as it is supposed to be written. Earlier in the manuscript, its citation was incomplete.

Reference (11) [Agizew T, Boyd R, Ndwapi N, Auld A, Basotli J, Nyirenda S, et al. Peripheral clinic versus centralized laboratory-based Xpert MTB/RIF performance: Experience gained from a pragmatic, stepped-wedge trial in Botswana. PLoS One. 2017 Aug 1;12(8).] formerly reference (11) in the manuscript has been removed from this manuscript. This is because the information it carries does not strongly connect with the content matter mentioned in line 67-68. 

Reference 13 has been re-referenced. Earlier in the manuscript, its citation was incomplete

Reference 18 has been re-referenced. Earlier in the manuscript, its citation was incomplete

Reference 21 has been re-referenced. Earlier in the manuscript, its citation was incomplete.

All the other references have been crosschecked for the correctness and support of content in the manuscript and we feel they are fit to be cited.

---

## [Decision Letter · Decision Letter 3]

3 May 2021

Implementation of GeneXpert MTB/Rif proficiency testing program: A Case of the Uganda national tuberculosis reference laboratory/Supranational reference laboratory

PONE-D-20-20339R3

Dear Dr. kabugo,

We’re pleased to inform you that your manuscript has been judged scientifically suitable for publication and will be formally accepted for publication once it meets all outstanding technical requirements.

Kind regards,

Shampa Anupurba, MD

Academic Editor

PLOS ONE

Additional Editor Comments (optional):

Reviewers' comments:

Reviewer's Responses to Questions

**Comments to the Author**

1. If the authors have adequately addressed your comments raised in a previous round of review and you feel that this manuscript is now acceptable for publication, you may indicate that here to bypass the “Comments to the Author” section, enter your conflict of interest statement in the “Confidential to Editor” section, and submit your "Accept" recommendation.

Reviewer #2: All comments have been addressed

2. Is the manuscript technically sound, and do the data support the conclusions?

Reviewer #2: Yes

3. Has the statistical analysis been performed appropriately and rigorously? 

Reviewer #2: Yes

4. Have the authors made all data underlying the findings in their manuscript fully available?

Reviewer #2: Yes

5. Is the manuscript presented in an intelligible fashion and written in standard English?

Reviewer #2: Yes

6. Review Comments to the Author

Reviewer #2: (No Response)

7. PLOS authors have the option to publish the peer review history of their article (what does this mean?). If published, this will include your full peer review and any attached files.

Reviewer #2: No

---

## [Editor Report · Acceptance letter]

5 May 2021

PONE-D-20-20339R3 

Implementation of GeneXpert MTB/Rif proficiency testing program: A Case of the Uganda national tuberculosis reference laboratory/Supranational reference laboratory 

Dear Dr. Kabugo:

I'm pleased to inform you that your manuscript has been deemed suitable for publication in PLOS ONE. Congratulations! Your manuscript is now with our production department. 

Kind regards, 

on behalf of

Dr. Shampa Anupurba 

Academic Editor

PLOS ONE